# Prognostic impact of the distance from the anterior surface to tumor cells in pancreatoduodenectomy with neoadjuvant chemoradiotherapy for pancreatic ductal adenocarcinoma

Miki Usui[1], Katsunori Uchida[1,2]*, Aoi Hayasaki[3], Masashi Kishiwada[3], Shugo Mizuno[3], Masatoshi Watanabe[1]

1 Department of Oncologic Pathology, Mie University Graduate School of Medicine, Tsu, Mie, Japan, 2 Department of Pathology, Kansai Medical University, Hirakata, Osaka, Japan, 3 Department of Hepato-Biliary-Pancreatic and Transplant Surgery, Mie University School of Medicine, Tsu, Mie, Japan

* uchidkat@hirakata.kmu.ac.jp

## Abstract

### Purpose

Several reports have shown the importance of margins in pancreatoduodenectomy (PD) specimens; however, whether anterior surfaces are included as margins varies among reports. In this study, we aimed to examine the impact of the anterior surface on disease-free survival (DFS) and overall survival (OS).

### Method

In total, 98 patients who underwent PD after chemoradiotherapy for pancreatic ductal adenocarcinoma at Mie University Hospital between January 1, 2012, and December 31, 2019, were included. We investigated the prognostic impact of the distance from the anterior surface to tumor cells on DFS and OS using a log-rank test. Multivariate analysis was performed using Cox proportional hazards analysis.

### Results

A significant difference in DFS and OS was observed up to a distance of 5 mm from the anterior surface of tumor cells. The multivariate analysis revealed that the distance from the anterior surface to tumor cells ($\leq 5$ mm) was an independent poor prognostic factor for DFS and OS.

### Conclusion

In patients with PD treated with neoadjuvant therapy, the distance from the anterior surface to tumor cells is an important assessment and should be included in the pathology report.

**Data Availability Statement:** All relevant data are within the manuscript and its Supporting information files.

**Funding:** The author(s) received no specific funding for this work.

**Competing interests:** The authors have declared that no competing interests exist.

## Introduction

Pancreatic ductal adenocarcinoma (PDAC) is a highly lethal disease, and the 5-year survival rate for all stages combined is 12%, according to the American Cancer Society [1]. Currently, the usefulness of neoadjuvant therapy has been reported not only for borderline disease but also for resectable disease [2]. The National Comprehensive Cancer Network (NCCN) guidelines indicate that neoadjuvant therapy is the first-line treatment for resectable or borderline resectable pancreatic cancer. Consequently, an increasing number of cases of chemoradiotherapy followed by surgical treatment is predicted in the future.

Several reports have examined prognostic factors in PD specimens, and the distance between the tumor cells and the margin is an independent prognostic factor [3–16]. Among these reports, some included the anterior surface as a margin, whereas others did not, suggesting that it is debatable whether the anterior surface should be considered a margin. The College of American Pathologists (CAP) protocol specifies that the presence of tumor cells within 1 mm of the resection margin is considered a positive margin; however, this rule does not apply to the anterior and posterior surfaces because they are not resection margins. It also states that reporting tumor invasion on the anterior and non-uncinate posterior surfaces is recommended but not required [17]. One report showed that invasion of the anterior surface was a poor prognostic factor [3]. In contrast, another study reported that tumor cells less than 1 mm away from the anterior surface had no prognostic impact [4]. However, these reports were based on patients who did not receive neoadjuvant therapy, and the prognostic relationship between the margin and prognosis in patients receiving neoadjuvant therapy is unclear.

We investigated the prognostic impact of the anterior surface in PD specimens treated with neoadjuvant therapy.

## Materials and methods

### Patients

This study was approved by the Clinical Research Ethics Review Committee of Mie University Hospital, waiving the need for informed consent due to the unfeasibility of obtaining individual consent from many dead patients and the retrospective nature of the study (reference number: H2018-073). The patients were given the opportunity to opt out of the study at any time, with details available on the Mie University School of Medicine website, ensuring adherence to the ethical standards outlined in the Declaration of Helsinki. Patient clinical and pathological data were extracted from the hospital's electronic medical records, anonymized, and stored in a database, maintaining strict confidentiality and being used solely for research purposes. The data cutoff was June 23, 2023. We extracted the clinical and pathological data of 134 patients who underwent pancreatectomy following chemoradiotherapy with gemcitabine and S-1 for pancreatic ductal adenocarcinoma at Mie University Hospital from January 1, 2012, to December 31, 2019 (Fig 1). Among these patients, one underwent total pancreatectomy, and 24 underwent distal pancreatectomy. We excluded five cases with no residual tumor, four cases where the tumor margin distance was indeterminable, and two cases of R2 resection.

### Pathological examination

The portal vein (PV)/superior mesenteric vein (SMV) margin, superior mesenteric artery (SMA) margin, and posterior surface of the PD specimens were inked according to the color code assigned by the surgeon. The anterior area without inking was designated as the anterior surface area. The specimens were fixed overnight in a 10% neutral-buffered formalin solution.

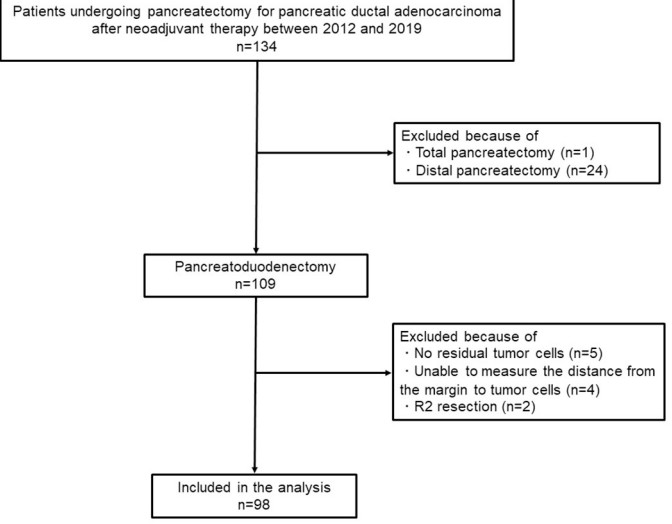

**Fig 1. Flow diagram of patient inclusion in the study.**

Pathologists sliced the PD specimen into 5-mm-thick slices in the axial plane, sampled all pancreatic tissues, and evaluated the pathological findings [18–20].

To validate the margins, the anterior surface and other margins (PV/SMV margin, SMA margin, and posterior surface) were validated as different factors. The distance from the anterior surface to the tumor cells and its impact on prognosis were examined at 0 mm, 0–X mm, and > X mm (X = 1, 2, 3, 4, 5, and 6). R1 was defined as the presence of tumor cells at a distance of 1 mm from any other margin except the anterior surface. Additionally, pancreatic transection, bile duct, and proximal/distal enteric margins were evaluated as independent factors.

## Statistical analysis

The Kaplan–Meier method was used to analyze disease-free survival (DFS) and overall survival (OS), with the surgery date as the starting point. The log-rank test was used to compare survival time distributions between the groups. A multivariate analysis was performed using the Cox proportional hazards model to identify significant prognostic factors. Statistical significance was set at $p < 0.05$. All statistical analyses were performed using EZR (Saitama Medical Center, Jichi Medical University, Saitama, Japan), a graphical user interface for R (The R Foundation for Statistical Computing, Vienna, Austria) [21].

## Results

### Patient characteristics

Of the 98 patients, 57 were male and 41 were female. The mean age at diagnosis was 66.6 years and ranged from 41 to 86 years. Sixty-one (62.2%) patients died of pancreatic cancer, 18 (18.4%) died of other causes, and 19 (19.4%) survived. Seventy (71.4%) patients experienced recurrence. The median time to recurrence was 13.5 months, and the median survival time was 25.5 months. Eighty-seven (88.7%) patients underwent resection of the PV or SMV. The relationship between each clinicopathological factor and DFS/OS is presented in Table 1. In

**Table 1. Pathological and clinical findings and outcomes.**

| variable | Number of patients (%) | Median DFS (months) | Log-rank p-value (DFS) | Median OS (months) | Log-rank p-value (OS) |
|---|---|---|---|---|---|
| Overall | 98 | 13.50 | | 25.49 | |
| **Age, years** | | | | | |
| <65 | 35 (35.7%) | 16.43 | 0.156 | 32.56 | 0.104 |
| ≧65 | 63 (64.3%) | 12.16 | | 22.90 | |
| **Gender** | | | | | |
| Male | 57 (58.2%) | 15.44 | 0.961 | 22.90 | 0.666 |
| Female | 41 (41.8%) | 12.32 | | 26.28 | |
| **Performance status** | | | | | |
| 0 | 68 (69.4%) | 14.85 | 0.371 | 26.87 | 0.041 |
| 1 | 23 (23.5%) | 12.68 | | 22.87 | |
| 2 | 6 (6.1%) | 8.16 | | 14.34 | |
| 3 | 1 (1.0%) | 8.25 | | 9.86 | |
| **Preoperative CA19-9** | | | | | |
| ≦37 | 66 (68.0%) | 21.91 | 0.048 | 29.47 | 0.278 |
| >37 | 31 (32.0%) | 15.87 | | 32.53 | |
| **Resectability after neoadjuvant therapy** | | | | | |
| R | 31 (31.6%) | 29.80 | 0.057 | 43.04 | 0.034 |
| BR-PV | 19 (19.4%) | 10.05 | | 20.99 | |
| BR-A | 18 (18.4%) | 11.14 | | 24.87 | |
| UR-LA | 30 (30.6%) | 11.24 | | 19.55 | |
| **Adjuvant therapy** | | | | | |
| No adjuvant | 17 (17.3%) | 7.72 | 0.092 | 7.79 | 0.017 |
| Adjuvant | 81 (82.7%) | 14.26 | | 29.50 | |
| **Tumor size** | | | | | |
| ≦20 | 35 (35.7%) | 27.66 | 0.007 | 36.44 | 0.055 |
| >20 | 63 (64.3%) | 11.14 | | 22.74 | |
| **Tumor grade** | | | | | |
| Well | 31 (31.6%) | 20.99 | 0.467 | 26.87 | 0.689 |
| Moderate | 25 (25.5%) | 9.86 | | 21.39 | |
| Poor | 42 (42.9%) | 13.04 | | 29.50 | |
| **Perineural invasion** | | | | | |
| Negative | 38 (38.9%) | 29.21 | 0.003 | 38.41 | 0.013 |
| Positive | 60 (61.2%) | 11.99 | | 22.74 | |
| **Lymphovascular invasion** | | | | | |
| Negative | 76 (77.6%) | 16.49 | 0.035 | 26.87 | 0.234 |
| Positive | 22 (22.4%) | 8.79 | | 16.99 | |
| **Extra-pancreatic nerve plexus invasion** | | | | | |
| Negative | 63 (64.3%) | 16.56 | 0.021 | 35.32 | 0.011 |
| Positive | 33 (35.7%) | 11.83 | | 22.77 | |
| **Grading of histological response [a]** | | | | | |
| 1 | 28 (28.6%) | 15.21 | 0.037 | 35.91 | 0.032 |
| 2 | 65 (66.3%) | 11.83 | | 21.78 | |
| 3 | 5 (5.1%) | 29.80 | | 34.60 | |
| **pT-stage [b]** | | | | | |
| 1 | 39 (39.8%) | 27.86 | 0.0011 | 41.43 | 0.054 |
| 2 | 51 (52.0%) | 10.05 | | 22.21 | |

*(Continued)*

**Table 1.** (Continued)

| variable | Number of patients (%) | Median DFS (months) | Log-rank p-value (DFS) | Median OS (months) | Log-rank p-value (OS) |
|---|---|---|---|---|---|
| **3** | 7 (7.1%) | 4.57 | | 12.71 | |
| **4** | 1 (1.0%) | 19.81 | | 32.30 | |
| **pN-stage** [b] | | | | | |
| **0** | 67 (68.4%) | 15.44 | 0.603 | 26.87 | 0.717 |
| **1** | 21 (21.4%) | 12.94 | | 25.82 | |
| **2** | 10 (10.2%) | 4.22 | | 12.63 | |
| **Margin status** | | | | | |
| **R0 (>1 mm)** | 64 (65.3%) | 15.93 | 0.105 | 26.58 | 0.371 |
| **R1 (≦1 mm)** | 34 (34.7%) | 12.07 | | 22.90 | |

Abbreviation: DFS: disease-free survival; DSS: disease-specific survival; OS: overall survival; R: resectable;

BR-PV: borderline resectable-portal vein; BR-A: borderline resectable-artery; UR-LA: unresectable-locally advanced.

[a] College of American Pathologists system.

[b] Union for International Cancer Control (UICC) TNM Classification of Malignant Tumors (ed 8), 2018.

one patient, preoperative serum CA19-9 levels were not measured, and in two patients, histological evaluation of extra-pancreatic nerve plexus invasion was impossible.

## Impact of the distance from the margin to the tumor cells

Log-rank tests showed a statistically significant difference between groups with a distance from the anterior surface to the tumor cells of X mm or less and greater than X mm for X = 1, 2, 3, 4, and 5. However, there was a significant difference in DFS (p = 0.027) but not in OS

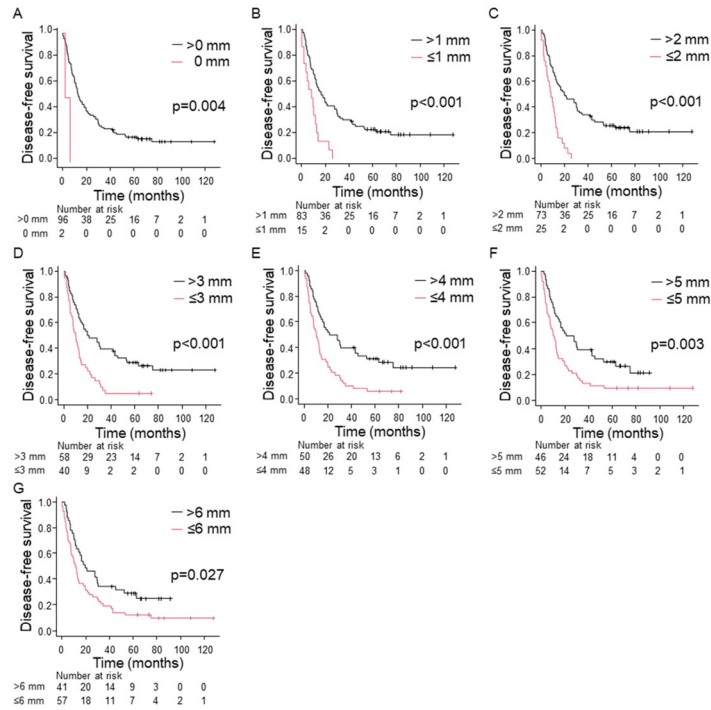

**Fig 2. Kaplan–Meier curves for 98 patients stratified into two groups by the distance from the anterior surface to tumor cells for DFS.** (A) 0 mm, (B) 1 mm, (c) 2 mm, (D) 3 mm, (E) 4 mm, (F) 5mm, (G) 6 mm.

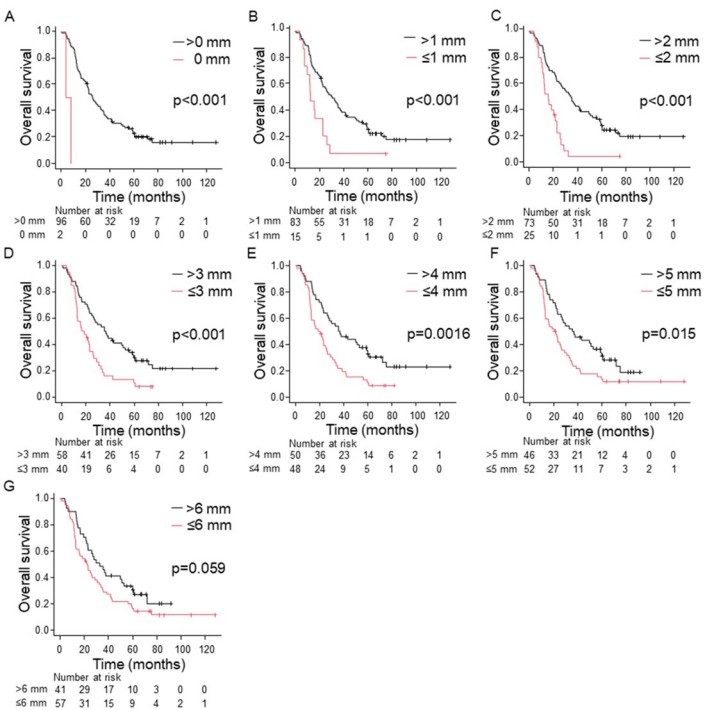

**Fig 3. Kaplan–Meier curves for 98 patients stratified into two groups by the distance from the anterior surface to tumor cells for OS.** (A) 0 mm, (B) 1 mm, (c) 2 mm, (D) 3 mm, (E) 4 mm, (F) 5mm, (G) 6 mm.

(p = 0.059) at X = 6 (Figs 2 and 3); when X ≤ 5 mm, significant differences in DFS and OS were shown between the groups. In contrast, margins other than the anterior surface (R1) were not prognostic for DFS or OS (p > 0.05) (Table 1). No tumor cells were within 1 mm of the pancreatic transection, bile duct, or proximal/distal enteric margins. Multivariate analysis also showed that tumor cells within X mm of the anterior surface were an independent poor prognostic factor for DFS and OS at X = 1, 2, 3, 4, and 5 (Table 2).

## Discussion

Several studies have examined the relationship between the distance from the margin to tumor cells and the prognosis of PD specimens. However, there are limited data on the prognostic impact of the distance between the anterior surface and tumor cells and the validation of PD specimens treated with neoadjuvant therapy. We showed that tumor cells within 5 mm of the anterior surface were associated with a poorer OS prognosis than those in the >5 mm group. In contrast, tumor cells within 1 mm of the margin, excluding the anterior surface (R1), did not affect DFS or OS.

Several reports have examined the distance between the margin and tumor cells and the prognosis in PD specimens without neoadjuvant therapy [3–16]. Some of these reports have not validated the anterior surfaces, likely because they were not resection margins [5, 6]. In addition, some reports did not describe the margins subjected to validation in detail [7, 8]. In these reports, depending on the study, the range of distances from the margin to the tumor cells that affected DFS and OS was from 0 to 2 mm. This may be caused by the differences between studies in the targeted margins, the distance defined as "a positive margin," whether or not inking was used, the extent of sampling, and so on. Few studies have examined individual margins. Regarding the distance between the anterior surface and tumor

**Table 2. Multivariate analysis.**

| Variable | DFS | | | OS | | |
|---|---|---|---|---|---|---|
| | HR | 95% CI | p-value | HR | 95% CI | p-value |
| **0 mm** | | | | | | |
| Resectability after neoadjuvant therapy, BR-PV+BR-A+UR-LA | 1.535 | 0.904–2.608 | 0.113 | 1.576 | 0.922–2.693 | 0.097 |
| Tumor size >20 mm | 1.305 | 0.747–2.280 | 0.349 | 1.116 | 0.628–1.983 | 0.709 |
| Perineural invasion, positive | 1.496 | 0.816–2.745 | 0.193 | 1.389 | 0.756–2.554 | 0.290 |
| Lymphovascular invasion, positive | 1.551 | 0.899–2.676 | 0.115 | 1.366 | 0.794–2.351 | 0.260 |
| Extra-pancreatic nerve plexus invasion, positive | 1.004 | 0.579–1.741 | 0.989 | 1.163 | 0.665–2.033 | 0.598 |
| Grading of histological response, Score 3 | 1.626 | 0.917–2.885 | 0.096 | 1.848 | 1.039–3.286 | 0.037 |
| Distance from the anterior surface to the tumor cells: 0 mm | 8.761 | 1.884–40.740 | 0.006 | 31.70 | 5.886–170.800 | p<0.001 |
| **1.0 mm** | | | | | | |
| Resectability after neoadjuvant therapy, BR-PV+BR-A+UR-LA | 1.419 | 0.825–2.439 | 0.206 | 1.481 | 0.856–2.561 | 0.160 |
| Tumor size >20 mm | 1.331 | 0.766–2.315 | 0.311 | 1.179 | 0.666–2.088 | 0.572 |
| Perineural invasion, positive | 1.495 | 0.810–2.758 | 0.199 | 1.342 | 0.728–2.472 | 0.346 |
| Lymphovascular invasion, positive | 1.720 | 0.985–3.002 | 0.056 | 1.289 | 0.750–2.216 | 0.358 |
| Extra-pancreatic nerve plexus invasion, positive | 0.844 | 0.475–1.502 | 0.565 | 0.941 | 0.520–1.702 | 0.841 |
| Grading of histological response, Score 3 | 1.496 | 0.847–2.643 | 0.165 | 1.765 | 1.002–3.110 | 0.049 |
| Distance from the anterior surface to the tumor cells: ≤1 mm | 2.765 | 1.422–5.379 | 0.003 | 2.966 | 1.512–5.820 | 0.002 |
| **2.0 mm** | | | | | | |
| Resectability after neoadjuvant therapy, BR-PV+BR-A+UR-LA | 1.480 | 0.870–2.516 | 0.148 | 1.492 | 0.870–2.559 | 0.147 |
| Tumor size, >20 mm | 1.213 | 0.689–2.134 | 0.503 | 1.056 | 0.591–1.889 | 0.854 |
| Perineural invasion, positive | 1.516 | 0.823–2.791 | 0.182 | 1.438 | 0.783–2.639 | 0.241 |
| Lymphovascular invasion, positive | 1.758 | 1.015–3.047 | 0.044 | 1.410 | 0.822–2.421 | 0.212 |
| Extra-pancreatic nerve plexus invasion, positive | 0.751 | 0.415–1.359 | 0.344 | 0.828 | 0.448–1.530 | 0.547 |
| Grading of histological response, Score 3 | 1.407 | 0.793–2.498 | 0.243 | 1.605 | 0.908–2.840 | 0.104 |
| Distance from the anterior surface to the tumor cells: ≤2 mm | 2.880 | 1.585–5.234 | p<0.001 | 3.078 | 1.631–5.809 | p<0.001 |
| **3.0 mm** | | | | | | |
| Resectability after neoadjuvant therapy, BR-PV+BR-A+UR-LA | 2.225 | 1.277–3.875 | 0.005 | 2.171 | 1.245–3.786 | 0.006 |
| Tumor size, >20 mm | 1.220 | 0.699–2.131 | 0.484 | 1.070 | 0.602–1.902 | 0.817 |
| Perineural invasion, positive | 1.236 | 0.670–2.283 | 0.498 | 1.132 | 0.617–2.079 | 0.689 |
| Lymphovascular invasion, positive | 1.654 | 0.963–2.839 | 0.068 | 1.367 | 0.799–2.338 | 0.253 |
| Extra-pancreatic nerve plexus invasion, positive | 0.794 | 0.446–1.412 | 0.432 | 0.948 | 0.533–1.686 | 0.855 |
| Grading of histological response, Score 3 | 1.563 | 0.877–2.785 | 0.130 | 1.704 | 0.961–3.024 | 0.068 |
| Distance from the anterior surface to the tumor cells: ≤3 mm | 2.650 | 1.561–4.497 | p<0.001 | 2.468 | 1.452–4.195 | p<0.001 |
| **4.0 mm** | | | | | | |
| Resectability after neoadjuvant therapy, BR-PV+BR-A+UR-LA | 2.152 | 1.251–3.700 | 0.006 | 2.113 | 1.226–3.640 | 0.007 |
| Tumor size, >20 mm | 1.072 | 0.613–1.875 | 0.806 | 0.944 | 0.530–1.683 | 0.845 |
| Perineural invasion, positive | 1.579 | 0.871–2.861 | 0.132 | 1.349 | 0.750–2.428 | 0.318 |
| Lymphovascular invasion, positive | 1.845 | 1.063–3.202 | 0.029 | 1.493 | 0.867–2.572 | 0.148 |
| Extra-pancreatic nerve plexus invasion, positive | 0.804 | 0.459–1.406 | 0.443 | 0.981 | 0.561–1.713 | 0.945 |
| Grading of histological response, Score 3 | 1.455 | 0.826–2.566 | 0.195 | 1.676 | 0.952–2.950 | 0.074 |
| Distance from the anterior surface to the tumor cells, ≤4 mm | 2.684 | 1.605–4.487 | p<0.001 | 2.498 | 1.496–4.169 | p<0.001 |
| **5.0 mm** | | | | | | |
| Resectability after neoadjuvant therapy, BR-PV+BR-A+UR-LA | 1.735 | 1.034–2.912 | 0.037 | 1.716 | 1.018–2.892 | 0.043 |
| Tumor size, >20 mm | 1.149 | 0.650–2.032 | 0.634 | 1.022 | 0.568–1.840 | 0.942 |
| Perineural invasion, positive | 1.448 | 0.795–2.637 | 0.226 | 1.291 | 0.710–2.348 | 0.403 |
| Lymphovascular invasion, positive | 1.806 | 1.031–3.163 | 0.039 | 1.493 | 0.860–2.593 | 0.155 |
| Extra-pancreatic nerve plexus invasion, positive | 0.991 | 0.577–1.701 | 0.973 | 1.167 | 0.675–2.021 | 0.580 |

*(Continued)*

**Table 2.** (Continued)

| Variable | DFS | | | OS | | |
|---|---|---|---|---|---|---|
| | HR | 95% CI | p-value | HR | 95% CI | p-value |
| **Grading of histological response, Score 3** | 1.398 | 0.793–2.463 | 0.247 | 1.598 | 0.908–2.812 | 0.104 |
| **Distance from the anterior surface to the tumor cells ≤5 mm** | 1.964 | 1.211–3.187 | 0.006 | 1.789 | 1.105–2.895 | 0.018 |

Abbreviation: HR: hazard ratio; CI: confidence intervals.

cells, one report showed that invasion of the anterior surface is a poor prognostic factor for 3-year survival [3], whereas another report showed that the presence of tumor cells within 1 mm of the anterior surface is associated with poor DFS [9]. However, some reports have shown that tumor cells within 0–2 mm of the anterior surface did not affect OS [4, 10]. All these reports were based on patients who did not receive neoadjuvant therapy. In our study, the distance from the anterior surface to the tumor cells for better DFS and OS was longer than that in previous reports. This discrepancy may be due to neither of these reports providing a detailed description of their tissue sampling methods and potentially sampling less tissue compared to our method, which sampled all pancreatic tissue. In pancreatectomy specimens post-preoperative adjuvant therapy, validated studies with extensively sampled specimens showed a pathological complete response rate of 2.5% [22], whereas validated studies with fewer sampled specimens tended to report higher rates [23, 24]. This suggests that a greater number of samples increases the detection rate of tumor cells, which may also impact the assessment of the distance from margins to the tumor cells. Few studies have investigated the relationship between the margins and prognosis of PD specimens after neoadjuvant therapy. Although one report indicated that the presence of tumor cells within 1 mm of the margin was associated with poor DFS and OS, no details were provided regarding which margins were validated [7]. Another report showed that the presence of a tumor within 1 mm of any of the margins was associated with poor DFS and OS, targeting the superior mesenteric/medial, pancreatic neck, bile duct, proximal gastric or duodenal, and distal jejunal margins, anterior surface, and posterior surface, but this was not verified for each margin [11]. A report analyzing the prognostic significance of tumor cell distance from margins in total and distal pancreatectomy following neoadjuvant therapy revealed that in total pancreatectomy, the proximity of tumor cells to the SMA margin impacts OS, whereas the distance from other margins does not influence survival. In contrast, for distal pancreatectomy, the proximity of tumor cells to both the anterior and posterior surfaces does not affect survival [25]. Extensive tissue sampling was conducted for pathological examinations in this study. Patients undergoing either total or distal pancreatectomy generally present with fewer symptoms of biliary stenosis and typically have more advanced tumors at diagnosis compared to those undergoing pancreatoduodenectomy; therefore, our findings are not directly comparable. Moreover, in PDAC, the closeness of tumor cells to the anterior surface is not the sole determinant of OS. Furthermore, distinguishing between non-neoplastic ductal cells and intraepithelial and invasive carcinoma in pancreatectomy specimens after preoperative adjuvant therapy can be challenging. Accurate pathological assessment is essential, as diagnostic discrepancies among pathologists may influence study outcomes [26].

Several reports have been published on the prognostic impact of tumor cell distance from the margins in PD specimens. However, no definite conclusion has been reached because of the varying verification methods used in individual reports. The prognostic impact of the distance from an individual margin to tumor cells depends on the margin to which the tumor

cells are in proximity [3, 11, 13]; further studies using appropriate inking and sampling specimens are required.

## Limitations

Our study has some limitations. First, this was a retrospective study and did not correct for patient characteristics. Second, this study was performed at a single center, and the influence of random errors caused by a relatively small number of cases could not be ruled out. Therefore, a multicenter study with a larger number of patients is required.

## Conclusion

This study is the first to focus on the anterior surface as the distance from the tumor margin in PD specimens treated with neoadjuvant therapy. We showed that tumor cells within 5 mm of the anterior surface were a poor prognostic factor for recurrence and survival. Therefore, the pathological report should describe the distance from the anterior surface to the tumor cells.

## Supporting information

**S1 Table. Patient data used for this study.**
(XLSX)

## Acknowledgments

The authors thank the members of the Departments of Oncological Pathology and Hepato-Biliary-Pancreatic and Transplant Surgery at Mie University for their valuable discussions and support.

## Author Contributions

**Conceptualization:** Miki Usui, Katsunori Uchida.

**Data curation:** Miki Usui, Katsunori Uchida, Aoi Hayasaki.

**Formal analysis:** Miki Usui.

**Investigation:** Miki Usui, Katsunori Uchida.

**Project administration:** Aoi Hayasaki, Masashi Kishiwada, Shugo Mizuno.

**Resources:** Aoi Hayasaki, Masashi Kishiwada, Shugo Mizuno.

**Supervision:** Katsunori Uchida, Masatoshi Watanabe.

**Visualization:** Miki Usui.

**Writing – original draft:** Miki Usui.

**Writing – review & editing:** Katsunori Uchida.

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
