## [Decision Letter · Decision Letter 0]

16 Jun 2024

PONE-D-24-08272Prognostic impact of the distance from the anterior surface to tumor cells in pancreatoduodenectomy with neoadjuvant chemoradiotherapy for pancreatic ductal adenocarcinomaPLOS ONE

Dear Dr. UCHIDA,

Thank you for submitting your manuscript to PLOS ONE. After careful consideration, we feel that it has merit but does not fully meet PLOS ONE’s publication criteria as it currently stands. Therefore, we invite you to submit a revised version of the manuscript that addresses the points raised during the review process.

We look forward to receiving your revised manuscript.

Kind regards,

Paolo Aurello

Academic Editor

PLOS ONE

Journal Requirements:

2. PLOS requires an ORCID iD for the corresponding author in Editorial Manager on papers submitted after December 6th, 2016. Please ensure that you have an ORCID iD and that it is validated in Editorial Manager. To do this, go to ‘Update my Information’ (in the upper left-hand corner of the main menu), and click on the Fetch/Validate link next to the ORCID field. This will take you to the ORCID site and allow you to create a new iD or authenticate a pre-existing iD in Editorial Manager. Please see the following video for instructions on linking an ORCID iD to your Editorial Manager account: https://www.youtube.com/watch?v=_xcclfuvtxQ".

Reviewers' comments:

Reviewer's Responses to Questions

**Comments to the Author**

1. Is the manuscript technically sound, and do the data support the conclusions?

Reviewer #1: Partly

Reviewer #2: Yes

2. Has the statistical analysis been performed appropriately and rigorously? 

Reviewer #1: Yes

Reviewer #2: Yes

3. Have the authors made all data underlying the findings in their manuscript fully available?

Reviewer #1: Yes

Reviewer #2: Yes

4. Is the manuscript presented in an intelligible fashion and written in standard English?

Reviewer #1: Yes

Reviewer #2: Yes

5. Review Comments to the Author

Reviewer #1: Pancreatic ductal adenocarcinoma has a 5-year survival rate of 12% for all stages of disease, it is therefore important to assess the efficacy of neoadjuvant therapy for such tumors, very carefully. Whilst the distance between tumor cells and pancreatoduodenectomy (PD) resection margin is an accepted prognostic factor, the significance of distance from the anterior or posterior surface is currently debated. This study aims to assess the prognostic impact of the distance from the anterior surface in PD specimens that underwent neoadjuvant therapy, and to provide insights for future reporting guidelines to include the distance of tumor cells from the anterior surface.

The researchers gathered data from 134 individuals who had undergone pancreatectomy surgery following chemoradiotherapy at Mie University Hospital between 2012 and 2019. Of those, 98 eligible patients, were included in the analysis.

The authors performed multi variate analysis and and rank-sum tests that produced a significant p-value that may indicate an association with poor OS and DFS, when tumor cells are found within 5mm of the anterior surface, they also acknowledged the limitations of the study, being a retrospective study that was performed in a single center.

while the study informs the need for further investigation, however, it does not seem sufficient to inform standards for reporting guidelines.

The authors do not provide a clear explanation of the biological reasoning behind their conclusions, and the challenges of assessing post-NAT specimens, however, they do mention that "In our study, the distance from the anterior surface to the tumor cells for better DFS and OS was longer than that in previous reports. This could have been caused by neoadjuvant therapy; however, this could not be clarified" [1]

The authors should cite and consider recent publications on the prognostic value of margin clearance [2]

References

Verbeke, C., Löhr, M., Karlsson, J. S., & Del Chiaro, M. (2015). Pathology reporting of pancreatic cancer following neoadjuvant therapy: challenges and uncertainties. Cancer treatment reviews, 41(1), 17-26.

Aaquist, T., Fristrup, C. W., Hasselby, J. P., Hamilton-Dutoit, S., Eld, M., Pfeiffer, P., ... & Detlefsen, S. (2024). Prognostic value of margin clearance in total and distal pancreatectomy specimens with pancreatic ductal adenocarcinoma in a Danish population-based nationwide study. Pathology-Research and Practice, 254, 155077.

Reviewer #2: A patient data is missing is very important for the practice of general surgery. The manuscript is well written. The data used are secondary and the institutional ethical approval is appropriate. Exclusion criteria are well stated. There is a patient data missing in the CA-19-9 result. Please cross-check.

6. PLOS authors have the option to publish the peer review history of their article (what does this mean?). If published, this will include your full peer review and any attached files.

Reviewer #1: No

Reviewer #2: **Yes: **PROF. LUKMAN OLAJIDE ABDUR-RAHMAN

---

## [Author Response · Author response to Decision Letter 0]

12 Jul 2024

July 7th 2024

Paolo Aurello

Academic Editor

PLOS ONE

Response to Reviewer #1: 

Thank you very much for reviewing our manuscript and offering valuable suggestions.

We have addressed your comments with point-by-point responses and revised the manuscript accordingly.

Comments 1 and 2: The authors do not provide a clear explanation of the biological reasoning behind their conclusions, and the challenges of assessing post-NAT specimens, however, they do mention that "In our study, the distance from the anterior surface to the tumor cells for better DFS and OS was longer than that in previous reports. This could have been caused by neoadjuvant therapy; however, this could not be clarified" [1]

The authors should cite and consider recent publications on the prognostic value of margin clearance [2]

Response: We appreciate the helpful suggestion. Following the reviewer's comment, we have compared our conclusions with those of recent reports on the prognostic value of margin clearance. Many of these reports did not clearly describe the procedure for sampling, and, likely, the search was not adequate. We have added the following to the Discussion (p.13, lines 144-151 and p13, lines 157-p14, lines 170): 

Regarding the longer distance from the anterior surface to tumor cells required to improve DFS and OS than previously reported, we have highlighted the method of tissue sampling (p.13, lines 144-151) and discrepancies in diagnosis between pathologists (p13, lines 157-p14, lines 170).

“This discrepancy may be due to neither of these reports providing a detailed description of their tissue sampling methods and potentially sampling less tissue compared to our method, which sampled all pancreatic tissue. In pancreatectomy specimens post-preoperative adjuvant therapy, validated studies with extensively sampled specimens showed a pathological complete response rate of 2.5% [22], whereas validated studies with fewer sampled specimens tended to report higher rates [23–24]. This suggests that a greater number of samples increases the detection rate of tumor cells, which may also impact the assessment of the distance from margins to the tumor cells.”

“A report analyzing the prognostic significance of tumor cell distance from margins in total and distal pancreatectomy following neoadjuvant therapy revealed that in total pancreatectomy, the proximity of tumor cells to the SMA margin impacts OS, whereas the distance from other margins does not influence survival. In contrast, for distal pancreatectomy, the proximity of tumor cells to both the anterior and posterior surfaces does not affect survival [25]. Extensive tissue sampling was conducted for pathological examinations in this study. Patients undergoing either total or distal pancreatectomy generally present with fewer symptoms of biliary stenosis and typically have more advanced tumors at diagnosis compared to those undergoing pancreatoduodenectomy; therefore, our findings are not directly comparable. Moreover, in PDAC, the closeness of tumor cells to the anterior surface is not the sole determinant of OS. Furthermore, distinguishing between non-neoplastic ductal cells and intraepithelial and invasive carcinoma in pancreatectomy specimens after preoperative adjuvant therapy can be challenging. Accurate pathological assessment is essential, as diagnostic discrepancies among pathologists may influence study outcomes [26].”

Response to Reviewer #2: 

Thank you very much for reviewing our manuscript and offering valuable suggestions.

We have addressed your comments with point-by-point responses and revised the manuscript accordingly.

Comment: A patient data is missing is very important for the practice of general surgery. The manuscript is well written. The data used are secondary and the institutional ethical approval is appropriate. Exclusion criteria are well stated. There is a patient data missing in the CA-19-9 result. Please cross-check.

Response: We appreciate the helpful suggestion. Following the reviewer's comment, we can confirm that preoperative serum CA19-9 levels had not been measured in one patient; we have changed the following text in the Result (p.5, lines 103-105)

Histological evaluation of extra-pancreatic nerve plexus invasion was impossible in two patients.

In one patient, preoperative serum CA19-9 levels were not measured, and in two patients, histological evaluation of extra-pancreatic nerve plexus invasion was impossible.

Thank you again for your comments on our manuscript. I trust that the revised manuscript now meets the standards for publication in PLOS ONE.

Sincerely,

Katsunori Uchida 

Department of Pathology, Kansai Medical University, Osaka, Japan

2-3-1 Shinmachi, Hirakata, Osaka, 573-1191, Japan

Phone number: +81-72-804-0101

Fax number: +81-72-804-2960

Email address: uchidkat@hirakata.kmu.ac.jp

---

## [Editor Report · Decision Letter 1]

15 Jul 2024

Prognostic impact of the distance from the anterior surface to tumor cells in pancreatoduodenectomy with neoadjuvant chemoradiotherapy for pancreatic ductal adenocarcinoma

PONE-D-24-08272R1

Dear Dr. KATSUNORI UCHIDA

We’re pleased to inform you that your manuscript has been judged scientifically suitable for publication and will be formally accepted for publication once it meets all outstanding technical requirements.

Kind regards,

Paolo Aurello

Academic Editor

PLOS ONE
---

## [Editor Report · Acceptance letter]

17 Jul 2024

PONE-D-24-08272R1 

PLOS ONE

Dear Dr. Uchida, 

I'm pleased to inform you that your manuscript has been deemed suitable for publication in PLOS ONE. Congratulations! Your manuscript is now being handed over to our production team.

Kind regards, 

on behalf of

Dr. Paolo Aurello 

Academic Editor

PLOS ONE